# Refractive Index Measurement Using SOI Photodiode with SP Antenna toward SOI CMOS-Compatible Integrated Optical Biosensor

**DOI:** 10.3390/s23020568

**Published:** 2023-01-04

**Authors:** Hiroaki Satoh, Koki Isogai, Shohei Iwata, Taiki Aso, Ryosuke Hayashi, Shu Takeuchi, Hiroshi Inokawa

**Affiliations:** 1Research Institute of Electronics, Shizuoka University, Hamamatsu 432-8011, Japan; 2Department of Engineering, Graduate School of Integrated Science and Technology, Shizuoka University, Hamamatsu 432-8561, Japan

**Keywords:** optical biosensor, sensor integration, refractive index measurement, label-free biosensing, SOI photodiode, surface plasmon antenna, FDTD method

## Abstract

This paper proposes a new optical biosensor composed of a silicon-on-insulator (SOI) p–n junction photodiode (PD) with a surface plasmon (SP) antenna. When the phase-matching condition between two lateral wavelengths of the diffracted light from the SP antenna and the waveguiding mode in the SOI PD is satisfied, we observe sharp peaks in the spectroscopic light sensitivity. Since the peak wavelength depends on the RI change around the SP antenna corresponding to the phase-matching condition, the SOI PDs with an SP antenna can be applied to the optical biosensor. The RI detection limit is evaluated in the measurements with bulk solutions, and 1.11 × 10^−5^ RIU (refractive index unit) can be obtained, which is comparable to that of a surface plasmon resonance (SPR) sensor, which is well known as a representative optical biosensor. In addition, the response for intermolecular bonds is estimated by the electromagnetic simulations using the finite-difference time-domain (FDTD) method to clarify its ability to detect biomolecular interactions. The results of this paper will provide new ground for high-throughput label-free biosensing, since a large number of SOI PDs with an SP antenna can be easily integrated on a single chip via an SOI complementary metal-oxide-semiconductor (CMOS) fabrication process.

## 1. Introduction

Refractive index (RI) measurement has found a variety of applications in the fields of agriculture, chemistry, biology, medicine, etc. In particular, RI-based detection of trace biomaterial without relying on fluorescence label—often referred to as label-free biosensing—has been researched using optical sensors. Label-free biosensing has advantages such as high versatility for the detection of various materials and unnecessity of the laborious labeling processes [1]. There are many types of optical biosensors, including surface plasmon resonance (SPR)-based sensors, interferometer-based sensors, optical-ring resonator-based sensors, optical-fiber-based sensors, photonic-crystal-based sensors, etc. Among them, the SPR sensor is representative because low RI detection limits from 10^−5^ to 10^−8^ RIU (refractive index unit) could be achieved for bulk solutions [2,3,4,5,6,7]. The range of the RI detection limit allows sensors to measure the clear signal change due to the imperceptible RI change induced by biomolecular interactions. However, most optical biosensors require a complicated optical system because the light source and photodetector must be arranged separately from the sensor surface. Additionally, increasing the signal-to-noise ratio is one of the most important aspects of sensor development. Thus, a simpler optical system is desirable to minimize the optical losses between the light source and the sensor surface, and between the sensor surface and the photodetector. Another challenge has been the integration of a large number of sensors on a chip for high-throughput analysis, which contributes to the acceleration of the research progress of life science, including early detection of disease, optimization of drug discovery, etc.

As one candidate satisfying both a simple optical system and high-density sensor integration, we introduce our proposed silicon-on-insulator (SOI) p–n junction photodiode (PD) with a gold (Au) line-and-space (L/S) SP antenna [8,9,10]. The Au SP antenna is used as a material to immobilize the biomolecules similar to SPR sensors. The PD has unique and advantageous features for RI measurement. The first is that a sharp peak can be observed in the spectroscopic light sensitivity when the phase-matching condition is satisfied between two lateral wavelengths of the diffracted light from the SP antenna and the waveguiding mode in SOI PD. Additionally, we have investigated the dependence of spectroscopic light sensitivity on incident angle to the PD surface [11,12,13,14] and demonstrated that the observed peak wavelengths were modulated by the phase difference in incident plane wave between two adjacent lines of SP antenna. Since the phase difference depends on the RI of the incident-side medium, the peak wavelengths are further modulated. As a result, SOI PDs with an SP antenna can sense the RI change around the SP antenna by observing the shift in peak wavelength. In the structure of the SOI PD with an SP antenna, the sensor part immobilizing the biomolecules and the photodetector part measuring the response to the RI change around the SP antenna are unified. The optical system is much simpler compared with the other optical biosensor, because the measurement can be performed by arranging the light irradiation system and the SOI PD with an SP antenna without the external photodetector. Additionally, the size of the PD is several tens of micrometers, which is about one-thousandth of an SPR sensor. A large number of sensors can be integrated with signal amplifiers and operation circuits in a single chip via an SOI complementary metal-oxide-semiconductor (CMOS)-fabrication process.

In this paper, the RI measurements using the SOI PD with an SP antenna are experimentally demonstrated, and the RI detection limit of our PD is evaluated. Changes in RI around the SP antenna are measured by using sucrose solutions with various concentrations as analyte. Additionally, it will be shown that the estimated detection limit to RI is comparable to the typical values of the SPR sensor. Furthermore, its ability to detect biomolecular interactions is estimated from the results of the electromagnetic simulations using the finite-difference time-domain (FDTD) method. In contrast with the other RI-based biosensors, the optical system of our sensor is much simpler, the sizes are small, and it can be integrated with CMOS circuits. The results may open up new horizons for high-throughput fluorescence-label-free biosensing.

## 2. SOI PD with SP Antenna

### 2.1. Device Structure

Figure 1 shows the device structure of a single SOI p–n junction PD with a Au SP antenna. Since the Au SP antenna is composed of periodic line-and-space (L/S) grating and a surrounding frame, this antenna can work not only as diffraction grating, but also as a gate electrode. Combined with the substrate electrode, the QE can be maximized by controlling the extent and position of the depletion region in SOI. The definitions of light incident angle θ and incident polarization with respect to the grating direction are also illustrated in the same figure. The PD has unique characteristics for the light selectivities of wavelength, incident angle, and polarization. These originate from the coupling efficiency between the diffracted light from Au L/S-type SP antenna and the laterally guided wave (in *x*-direction) based on the eigenmode in the slab waveguide consisting of the SOI sandwiched by silicon dioxide (SiO_2_). When the phase-matching condition between the wavelengths in *x*-direction of these lights is satisfied, the light-absorption efficiency in SOI can be enhanced, resulting in higher QEs with a sharp peak in wavelength and incident angle [8,9,10,11,12,13,14]. Since the wavelength in *x*-direction of the diffracted light is modulated by the RI of the medium around the SP antenna, it becomes possible for our proposed PD to sense an imperceptible change in the RI corresponding to biomolecular interactions. This physical mechanism is explained in Section 3.

Figure 2 shows various top views of microscopic images of a fabricated device. The integrated SOI PDs with an SP antenna and diodes are arranged as our prototype of integrated optical biosensors, as shown in Figure 2a. For RI and temperature measurements, 4 × 7 SOI PDs (column A to C and E to H) with SP antenna and 4 × 1 diodes (column D) are integrated, respectively, in a fabricated chip. The diodes are used for temperature compensation of the analyte in RI measurements. The SOI p^−^ area of diodes must be shaded by the metal cover to disable any light responses, which consists of lower Ti and upper Au films with a thicknesses of 20 nm and 300 nm, respectively. The 300 nm-thick Au can protect any light irradiations to the PD body because the thickness is sufficiently larger than Au skin depth in the visible light range [15]. On the other hand, the p^−^ light-sensitive area of SOI PD is covered by a SiO_2_ and Au/Ti L/S grating-type SP antenna, as mentioned previously. Thus, the SOI PD with an SP antenna works as a RI-based biosensor. A unit SOI PD with an SP antenna is shown in Figure 2b. Since we introduce the aqueous solutions of analyte around the SP antenna, the unexpected electrical short between devices through the solution must be avoided. Thus, the contact pads and the feed lines are covered by photoresist, but this photoresist around the SP antenna is removed by photolithography. SP antennas are completely exposed to sense the RI measurements for bulk solutions and the interactions of molecules immobilized on the surface of SP antenna. The scanning electron microscope (SEM) image of the SP antenna is shown in Figure 2c. The SP antenna is composed of a Au/Ti L/S grating structure. In this image, the grating period and line width are *p* = 300 nm and *w* = 150 nm, respectively. The direction of the electric field of incident light is polarized perpendicular to the L/S direction of the SP antenna.

### 2.2. Fabrication Process

The fabrication process is summarized here: (1) The SOI thickness of a commercial p-type SOI wafer is adjusted by thermal oxidations and SiO_2_ etchings. The designed SOI thickness at the fabrication completion is *t*_SOI_ = 100 nm. The buried oxide (BOX) thickness of this wafer is *t*_BOX_ = 200 nm. The initial impurity (boron) concentrations in the SOI and the substrate are both 1 × 10^15^ cm^−3^. (2) The SOI pattern for the isolation between devices is delineated by photolithography and Si etching. (3) Boron fluoride (BF_2_^+^) and arsenic (As^+^) ions are implanted to the anode and the cathode regions, respectively, of the SOI. (4) Gate oxide with the thickness of *t*_ox_ = 100 nm is formed by the thermal oxidation and SiO_2_ depositions, and the contact holes are opened. (5) SP antennas and contact pads with Au and titanium (Ti) are fabricated by electron-beam (EB) lithography, Ti and Au evaporations, and lift-off. The Au thicknesses for the SP antennas and contact pads are 100 nm and 300 nm, respectively. In order to obtain sufficient adhesion strength between SiO_2_ and Au, a thin Ti layer is inserted. The Ti thicknesses for SP antennas and contact pads are 5 nm and 20 nm, respectively. To compensate the temperature of the analyte, the temperature sensors of a simple p–n junction diode are arranged in the same chip. These diodes are covered by a Au layer with thickness of 300 nm, which is sufficiently greater than its skin depth, to disable any optical influences. This Au layer can be also used as a gate electrode of temperature sensors and fabricated simultaneously with contact pads. (6) Finally, the photoresist is coated and patterned as a protective film above the devices to avoid an unexpected electrical short through the aqueous solution for RI measurements. Only the SP antennas on PDs and the Au covers on diodes are exposed to sense the refractive index and the temperature of injected aqueous solutions, respectively.

### 2.3. PD Operation Condition

p–n junction PDs can detect light based on the photocurrent when a reverse bias is applied to the p–n junction. Since the photocurrent depends on the number of photogenerated electron–hole pairs in the depletion region and the drift velocities of the carriers [16], the depletion region should be expanded entirely in the p^−^ light-sensitive region underneath the SP antenna. In this paper, the cathode voltage, which is applied to the n^+^ region, is fixed at *V*_C_ = +1 V, and the anode electrode, which is connected to the p^+^ region, is grounded (*V*_A_ = 0). The gate voltage *V*_G_ and the substrate voltage *V*_SUB_ are adjusted to apply an appropriate static electric field into the PD body, so that the depletion region is formed entirely in a p^−^ light-sensitive region. Figure 3 shows the schematic models of different photocurrents depending on the depletion region and the measured PD current as a function of *V*_SUB_ for a gate voltage of *V*_G_ = 0 V. In the neutral condition with no biases to the gate and the substrate, the carrier distribution and the depletion region are expected as shown in the left figure of Figure 3a, resulting in a low photocurrent due to the small volume of the depletion region in the p^−^ light-sensitive area. On the other hand, when appropriate bias voltages are applied to the gate and the substrate electrodes, the volume of the depletion region and the photocurrent can be maximized, as shown on the right-hand side of Figure 3a. In this example, the top Si area at the gate oxide/SOI interface is inverted (“inv.”) by the positive bias voltage applied to the gate electrode, and the bottom Si area at the SOI/BOX interface is accumulated (“acc.”) by the negative bias voltage applied to the substrate electrode. When the top Si area is accumulated and the bottom Si area is inverted, the photocurrent can be also maximized. However, we will introduce aqueous solutions around the gate electrode (SP antenna) in RI measurements. Therefore, the PD current as a function of *V*_SUB_ was evaluated in Figure 3b, when no bias voltage was applied to the gate electrode (*V*_G_ = 0 V). The TM-polarized light with a wavelength of 685 nm was irradiated normal to the PD. The grating period of SP antenna is *p* = 340 nm. The maximized PD current region was observed from *V*_SUB_ = −20 V to −16 V. This result indicates that the SOI p^−^ light-sensitive area could be entirely depleted by the appropriate substrate voltage, even though no bias voltage was applied to the gate electrode. This result contributes to the desirable condition without any gate voltage in the RI measurements with aqueous solutions. It is confirmed that the dark currents lower than 1 pA were measured with the common voltage conditions, which are negligible with the photocurrents.

## 3. Principles of Refractive Index Measurement

### 3.1. Peak Wavelength Shifts in Spectroscopic Photocurrent (Single PD Method)

The principle for RI measurement using the SOI PD with an SP antenna is based on the phase-matching condition between the diffracted light from the L/S grating-type SP antenna and the guided mode in the SOI PD body. When the phase-matching condition is satisfied, the peak appears in spectroscopic photocurrent [11,12,13,14]. Considering the additional condition for the RI on the incident side, the modified phase-matching condition can be obtained, as shown in Figure 4. The phase difference Δ of the incident plane wave between two adjacent lines occurs only for oblique incidence, as follows:(1)$$\Delta =p\left(\frac{2\pi n}{\lambda }\right)\sin\theta$$
where *p* is the grating period of the SP antenna, *n* is the RI of incident side, and λ is the incident wavelength. Additionally, two solutions exist for the forward and the backward waves of the guided mode, as follows:(2)λgf=1 1p+nsinθλ 
(3)λgb=1 1p−nsinθλ 
where λ_gf_ and λ_gb_ are the propagation wavelengths of guided mode for the forward and backward waves, respectively.

Since the SOI layer (core with higher RI) is sandwiched by SiO_2_ (cladding with lower RI), the guided modes exist in *x*-direction. The propagation wavelength λ_g_ in SOI waveguide for the TM wave can be numerically analyzed using the following transcendental equation [17]:(4)$$\tan\left(\frac{ht_{\mathrm{core}}}{2}-m\frac{\pi }{2}\right)=\frac{n_{\mathrm{core}}^{2}}{n_{\mathrm{cladding}}^{2}}\cdot \frac{\sqrt{V^{2}-h^{2}t_{\mathrm{core}}^{2}}}{ht_{\mathrm{core}}}\;\;\;(m=0,\;1,\;2,\;\ldots )$$
where
$$h=\sqrt{{\left(\frac{2\pi n_{\mathrm{core}}}{\lambda }\right)}^{2}-{\left(\frac{2\pi }{\lambda_{g}}\right)}^{2}},\;\;\;\;V=\frac{2\pi }{\lambda }t_{s}\sqrt{n_{\mathrm{core}}^{2}-n_{\mathrm{cladding}}^{2}},$$

*t*_core_ is the SOI thickness and *n*_core_ and *n*_cladding_ are the RIs of Si and SiO_2_, respectively. Substituting λ_g_ analyzed from Equation (4) for λ_gf_ in Equation (2) or λ_gb_ in Equation (3), the peak wavelength λ in spectroscopic photocurrent can be expected. Thus, the RI can be sensed by the peak wavelength in the measured spectroscopic photocurrent of SOI PD with the SP antenna. This RI measurement method is called the single PD method.

Figure 5 shows the measured spectroscopic photocurrents of two RIs for normal and oblique incidences. We introduced the insulating liquid (bulk analyte) around the SP antenna to vary the RIs. The grating period of the SP antenna is *p* = 300 nm. When the incident light is normal to the PD (θ = 0), the peak shape and the peak wavelength are almost unchanged. In this case, one peak appears at the incident wavelength of 698 nm because its propagation wavelength based on Equation (4) matched the grating period (λ_g_ = *p* = 300 nm). The results prove that the peak wavelengths are independent of the RI *n* for normal incidence. On the other hand, the peak wavelength shifts for oblique incidences with the angle of θ = 10° or 20° are clearly observed due to the dependence of the phase difference Δ on the RI *n* around SP antenna. In actual RI measurements, an appropriate incident angle exists because there is a trade-off relationship between the peak photocurrent and the peak wavelength shift corresponding to the signal-to-noise (S/N) ratio and the RI sensitivity, respectively. Figure 6 shows the dependence of the peak wavelength difference [Δλ_p_(*n*) = λ_p2_(*n*) − λ_p1_(*n*)] on the RI, where λ_p1_(*n*) and λ_p2_(*n*) are the peak wavelengths for forward and backward waves, respectively. The theoretical results based on the phase-matching condition are also provided. The RI sensitivities are evaluated by the slopes. The measured values agree well with the theoretical ones. The results indicate that the modified phase-matching condition in Figure 4 can be applied to the RI measurement using the SOI PD with an SP antenna. Additionally, the RI sensitivities based on the peak wavelength difference normalized by the central wavelength are comparable with the one for the Si ring resonator-based biosensor [18], which has been successfully applied to RI measurements based on the resonant wavelength shifts. On the other hand, wavelength scanning is required to extract the peaks from spectroscopic photocurrent, and thus the measurement time is quite long. In order to realize rapid or real-time measurements, we propose the new single-wavelength dual PD method in Section 3.2.

### 3.2. Photocurrent Difference of Two PDs (Single-Wavelength Dual PD Method)

Since the single PD method measures the shift in peak wavelength in spectroscopic photocurrent depending on RI, wavelength scanning is required. Thus, the measurement time lengthens, and the spectrometer is required in the optical irradiation system. In order to resolve these issues, we propose a new method to realize an efficient RI measurement without wavelength scanning, which is called the single-wavelength dual PD method. Figure 7 shows the schematic of the optical system and the operation principle in the case of forward wave in the single-wavelength dual PD method. In this method, RIs can be evaluated by the differential photocurrents of two PDs with different periods (*p*_1_ and *p*_2_) at a designed wavelength. Since there is no need to analyze the spectroscopic characteristics, the laser diode (LD) or light-emitting diode (LED) with a single wavelength and high optical intensity can be used as light sources in the optical irradiation system. This method allows us to reduce the measurement time and improve the S/N ratio. After defining the incident wavelength and the initial RI *n*_0_, the incident angle and the combination of two grating periods (*p*_1_ and *p*_2_) are designed to coincide with the photocurrents of two PDs based on the measured spectroscopic photocurrents of SOI PDs with various grating periods. The peak wavelengths of two PDs are modulated to the shorter wavelength by the RI increase around the SP antenna from *n*_0_ to *n*_0_ + Δ*n*. Thus, the differential photocurrent increases. Note that the slope of the differential photocurrent for RI variation, which corresponds to RI sensitivity, is ideally maximized, if the cross point could be defined at the half maximums of the photocurrents of two PDs.

## 4. Results and Discussion

### 4.1. Sensitivity and Detection Limit for Refractive Index

Figure 8 shows the measurement circuit for the single-wavelength dual PD method. The laser light with the wavelength of 685 nm is TM-polarized by a half-wavelength plate, and the collimated beam created using the beam expander is irradiated to two PDs. Based on the phase-matching condition, the periods of SP antenna are *p*_1_ = 330 nm and *p*_2_ = 340 nm, and the incident angle is 16.8°. The light intensity of the laser diode is 48 mW/cm^2^, and it is modulated by the sinusoidal wave with a frequency of 500 Hz for lock-in detection to reduce the influences of the light-intensity fluctuations. The PD currents are converted to the voltages by a pre-amplifier with a sensitivity of 10 MV/A, and the output voltage is based on the differential signals of two PD currents in a lock-in amplifier. The temporal output voltage in the lock-in amplifier is recorded. This is called a sensorgram. 

In our measurements, the signal drifts in the measured sensorgram were observed due to the RI changes depending on the solution temperature. In order to compensate for the signal drift, the SOI p–n junction diode-based temperature sensors are used. The forward current *I*_F_ of the p–n junction diode obeys the following diode equation [16]:(5)$$I_{F}\approx \;\exp\left(\frac{qV}{\eta kT}\right)$$
where *q* and *V* are the elementary charge and the forward bias voltage, respectively. η, *k*, and *T* are the ideality factor of the fabricated diode, the Boltzmann constant, and the temperature, respectively. If a constant forward current is applied to the diode, the voltage difference between the anode and the cathode depends on the temperature around the diode body. We confirmed that the temperature sensitivity of 3.04 × 10^−3^ V/K in the fabricated SOI diode (temperature sensor) was obtained at the constant forward current of 1 nA, and the signal drift in the measured sensorgram could be eliminated by considering the RIs of aqueous solutions at the temperature of 20 °C [19]. This treatment is one of the most important advantages in the use of integrated circuits for biosensing, as there is no need to install the external temperature sensor in our measurement system. Additionally, the actual temperature of analytes can be measured by arranging the temperature sensor very close to the PDs.

In order to evaluate the RI detection limit of bulk analyte, the aqueous solutions are continuously introduced around the SP antenna by a pumping system with a sample injector. In this measurement, pure water and sucrose solutions with concentrations of 1% or lower based on the weight of sucrose divided by the one containing pure water are used as a buffer and the analytes, respectively. The RIs of these solutions corresponds to the range from *n* = 1.3330 to 1.3344 at the temperature of 20 °C.

Figure 9 shows the measured sensorgram of the output voltage in the RI measurement of sucrose solutions with various concentrations using the single-wavelength dual PD method. It can be observed that the signal levels are clearly varied due to RI changes depending on the concentrations of the sucrose solution. Since the baseline is defined for the RI of pure water (buffer), the signal levels return to the base line when the analytes are completely drained from the region around the SP antenna.

Figure 10 shows the calibration curve obtained from the measured sensorgram in Figure 11. The output voltage is linearly increased as the RI increases. The slope of this characteristic corresponds to the RI sensitivity. In this case, the RI sensitivity of 3.49 V/RIU (RIU: refractive index unit) is obtained. In addition, the noise level evaluated by the standard deviation of the fluctuation in the flat region for pure water is 38.7 μV. The RI detection limit of our sensor can be estimated from the RI sensitivity and the fluctuations of flat regions in the sensorgram as the following equation.
(6)$$\mathrm{RI}\;\mathrm{detection}\;\mathrm{limit}\;\left[\mathrm{RIU}\right]\;=\;\;\frac{\mathrm{Noise}\;\mathrm{level}\;\left[V\right]}{\mathrm{RI}\;\mathrm{sensitivity}\;\left[V/\mathrm{RIU}\right]}$$

As a result, a detection limit of 1.11 × 10^−5^ RIU can be attained. This value is comparable to that of the SPR sensor [1,2,3,4,5].

### 4.2. FDTD Estimation for Optical Response to Biomolecular Interactions

The optical response to biomolecular interactions in the single-wavelength dual PD method is estimated. The two-dimensional (2D) electromagnetic simulations based on the FDTD method are performed to investigate the absorption efficiencies in the SOI layer of two SOI PDs with an SP antenna, which is equivalent to the external QEs in the fabricated results. Figure 11 summarizes the structure and the conditions in the FDTD simulation. In order to save on computation costs for the consumption of time and memory storage, a unit periodic structure with the grating period *p* is assumed. The boundary conditions for computational space in the *y*- and *z*-directions are a periodic boundary with the constant-*k* method [20] and an absorbing boundary with the perfectly matched layer (PML) [21], respectively. The discretized interval and the time step are Δ*d* (=Δ*y* = Δ*z*) = 1 nm and Δ*t* = Δ*d*/(2 *c*_0_) = 1.67 × 10^−18^ s, respectively, where *c*_0_ is the light speed in a vacuum. The relative permittivities for SiO_2_ and vacuum are ε_r_ = 2.13 and 1, which are independent of the wavelength. The complex permittivity of Si is ε_r_ = 14.5 + *j*0.102 at the wavelength of 685 nm [22]. Au and Ti show negative permittivities in the visible-light range [15]. In this case, the electromagnetic fields can be appropriately computed without numerical divergences when Au and Ti are treated as dispersive media in the FDTD method [23]. In this paper, the Lorentz-Drude dispersion model [15] is used to express the complex permittivities of Au and Ti, and the piecewise-linear recursive-convolution (PLRC) method [24] is adopted to introduce the dispersive characteristics of the complex permittivities into the FDTD calculations. 

Figure 12 compares the simulated external QE in SOI equivalent to absorption efficiency and the measured QE as a function of the incident angle, when the buffer solution (pure water) with a RI of *n*_0_ = 1.3325 is introduced around the SP antenna. The simulated peak incident angles are in good agreement with the measured ones for both grating periods. Furthermore, the cross point between the simulated QEs for two grating periods also shows good agreement with the one between the measured QEs, which is located at an incident angle of 16.4°. These results illustrate that the simulated results have good accuracy for the estimation of optical response, and the optimizations of the optical irradiation system and the device structure. Similar to the spectroscopic QE, the peaks in the dependence of QE on the incident angle appear when the phase-matching condition is satisfied between the diffracted light from the SP antenna and the SOI waveguiding mode. The grating period of SP antenna and the SOI thickness depend primarily on the peak incident angle. Since the SP antenna was fabricated by EB lithography, the grating period can be accurately fabricated. Additionally, the measured SOI thickness of the fabricated device by the commercial thickness meter based on the spectroscopic reflectance was 99.8 nm.

Figure 13 shows the calibration curves obtained from the absorption efficiencies of PD1, PD2, and the differential one between two PDs for various RIs. It can be confirmed that the absorption efficiency increases linearly as the RI of analyte increases. Furthermore, the simulated RI sensitivity based on the differential QE is comparable to the measurements obtained. Therefore, the FDTD simulation can be adopted for the estimation of the optical response for biomolecular interactions of a single-wavelength dual PD method.

Before the FDTD simulation with biomolecular interactions, the thickness of the molecular layer immobilized on the flat Au surface is experimentally measured. Figure 14 shows the schematic of the molecular layer immobilized on Au surface and the measured molecular layer thicknesses by ellipsometer with the 632.7 nm light. Avidin, a protein found in egg white which has a strong affinity for biotin (also known as vitamin B7) [25,26], is selected as a model biomolecule. The procedure to immobilize the molecules on the Au surface is as follows:Cleaning the Au surface of the sample using the 1:3 mixture of hydrogen peroxide solution and sulfuric acid for 30 s.Dipping the sample in 1 mM cysteamine water solution for 30 min. Cysteamine is a type of thiol with an affinity for Au, and it provides an amino end group to facilitate the biotinylation reagent in the next step.Preparing a 50 mM biotin N-hydroxy-sulfosuccinimide ester (Biotin Sulfo-OSu) water solution, and then diluting it by 10 with phosphate-buffered saline (PBS) with a pH of about 7.4. The sample is dipped in the solution for 30 min to create a cysteamine-biotin complex.Preparing a 100 μg/mL water solution of avidin, and then diluting it by 10 with PBS. The sample is dipped in the solution for 30 min.At the end of each step, the sample is washed with pure water, dried by blowing nitrogen, and measured by ellipsometry to determine the thickness. As shown in Figure 14b, sequential attachment of the nanometer-scale molecules is observed.

In FDTD estimation, the molecular layer immobilized on the Au surface is expressed by the homogenously deposited layer on the Au surface. Figure 15 shows the schematic model of the deposited layer and the simulated QE change as a function of analyte thickness based on the thickness measured via ellipsometry. Considering the measurement situation, the PBS is continuously introduced around the SP antenna, and thus the RI of *n*_buffer_ = 1.3325 is assumed on the incident side. Additionally, the RI for the deposited layer (analyte) is fixed at *n*_analyte_ = 1.5, similar to the ellipsometry treatment. The change in QE with respect to analyte thickness from *t*_analyte_ = 0 to 5 nm is simulated for *p*_1_, *p*_2_, λ, and θ of 330 nm, 340 nm, 685 nm, and 16.4°. The differential QE change (PD2–PD1) up to 3 × 10^−3^ is attained, which is sufficiently large for the experimentally detectable QE change demonstrated in Figure 9 and Figure 10. Therefore, our RI-based biosensor based on a single-wavelength dual PD method can detect the biomolecular bonds with high sensitivity and contributes to high-throughput analyses using integrated circuits. 

## 5. Conclusions

Capability of the PD with a Au SP antenna for RI measurement was accessed, and the significant shift in the QE peak was observed due to the modified phase-matching condition with the RI of the incident side between the diffracted light from the SP antenna and the waveguiding mode in the Si layer for oblique incidence. Based on the single-wavelength dual PD method with differential lock-in detection of the modulated laser light, the RI detection limit for sucrose aqueous solution was 1.11 × 10^−5^ RIU, which is comparable to that of the SPR sensor. Considering the avidin–biotin complex as an example of a biomolecule, the effect of the nanometer scale deposition on the Au lines of the SP antenna is estimated by FDTD simulation. Sufficient change in QE, based on the experimentally detectable QE change, was obtained, and its biosensor prospects were clarified. Featuring the small device and the high RI sensitivity, the proposed PD may provide a new opportunity for biosensing techniques in terms of high-throughput analysis using SOI CMOS integrated circuits.

## Figures and Tables

**Figure 1 sensors-23-00568-f001:**
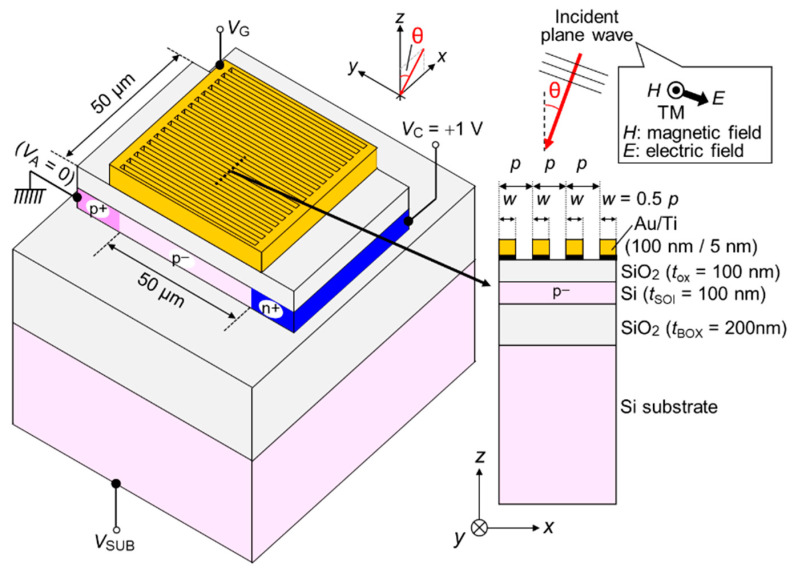
Bird’s-eye view and cross-sectional view of a single silicon-on-insulator photodiode (SOI PD) with surface plasmon (SP) antenna and definitions of incident polarization and incident angle θ. The light-sensitive area equivalent to p^−^ active area is 50 × 50 μm^2^. Gold (Au) SP antenna composed of line-and-space (L/S) structure and surrounding frame can work not only as a diffraction grating, but also as a gate electrode for higher quantum efficiency (QE) by controlling the extent and position of depletion region in SOI.

**Figure 2 sensors-23-00568-f002:**
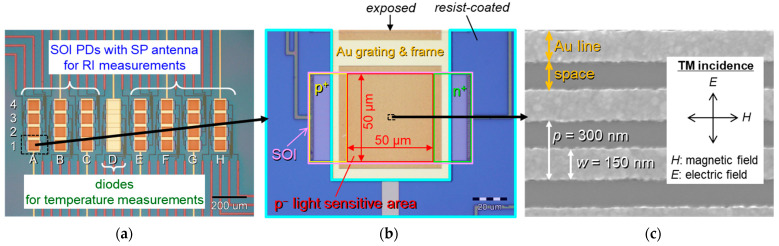
Top-views of microscopic images of fabricated devices. (**a**) Integrated silicon-on-insulator photodiodes (SOI PDs) with surface plasmon (SP) antenna and diodes on a chip. First, 4 × 7 PDs (column A to C and E to H) and 4 × 1 (column D) diodes are arranged for refractive index (RI) measurements and temperature measurements of analyte, respectively. (**b**) A single SOI p–n junction PD with SP antenna. The SP antenna is exposed to sense refractive index (RI), and the other area is protected by photo-resist to avoid unexpected electrical short through aqueous solution. (**c**) Scanning electron microscope (SEM) image of a gold (Au) line-and-space (L/S)-type SP antenna with the grating period of *p* = 300 nm. The inset is the definition of incident polarization to the grating direction.

**Figure 3 sensors-23-00568-f003:**
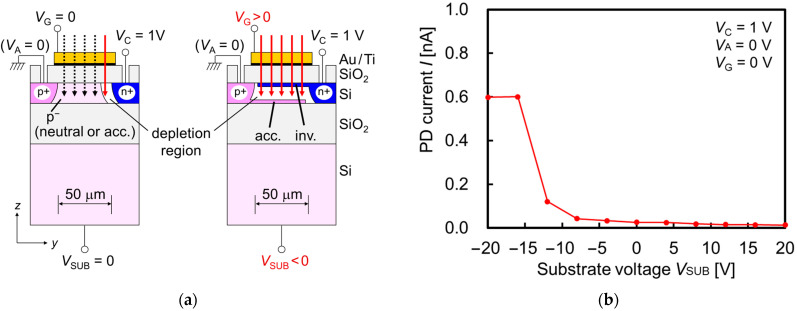
(**a**) Schematic models of different photocurrent depending on depletion region. Solid and dotted arrows indicate detectable and undetectable light rays, respectively, due to position and volume of depletion region. The “acc.” and “inv.” represent “accumulated” and “inverted”, respectively. (**b**) Measured photodiode (PD) current as a function of substrate voltage *V*_SUB_, when no bias voltage was applied to gate electrode (*V*_G_ = 0 V). Transverse magnetic (TM)-polarized light with the wavelength of 685 nm is irradiated normal to silicon-on-insulator (SOI) PD with surface plasmon (SP) antenna. The voltages of cathode and anode are fixed at *V*_C_ = 1 V and *V*_A_ = 0 V, respectively.

**Figure 4 sensors-23-00568-f004:**
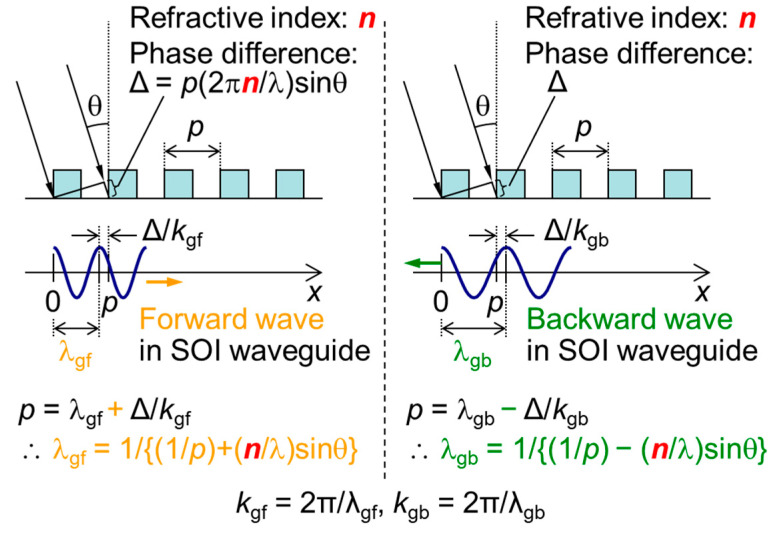
Schematic diagrams of phase-matching conditions between diffracted lights and forward and backward waves in the silicon-on-insulator (SOI) waveguide. *n* is refractive index (RI) of the incident side region. *k*_gf_ and *k*_gb_ are wavenumbers of the forward and backward waves, respectively. λ_gf_ and λ_gb_ are propagation wavelengths of the forward and backward waves, respectively.

**Figure 5 sensors-23-00568-f005:**
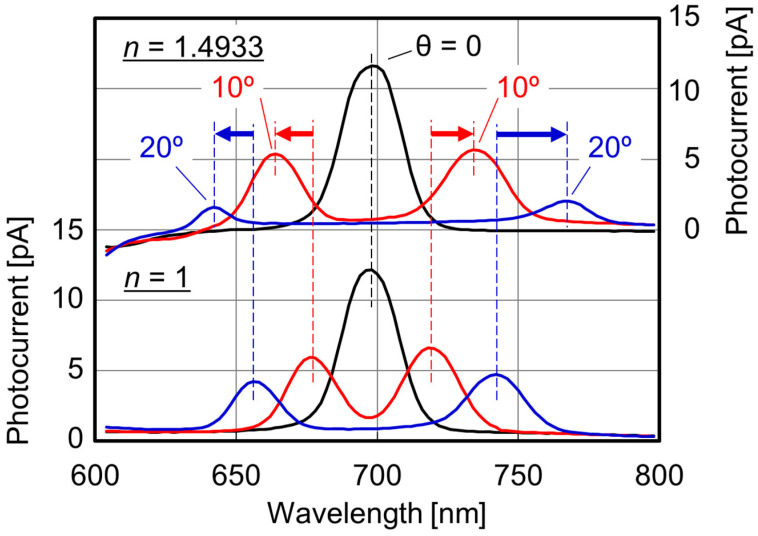
Measured spectroscopic photocurrent with various incident angle θ for two refractive indices. The period of surface plasmon (SP) antenna is *p* = 300 nm.

**Figure 6 sensors-23-00568-f006:**
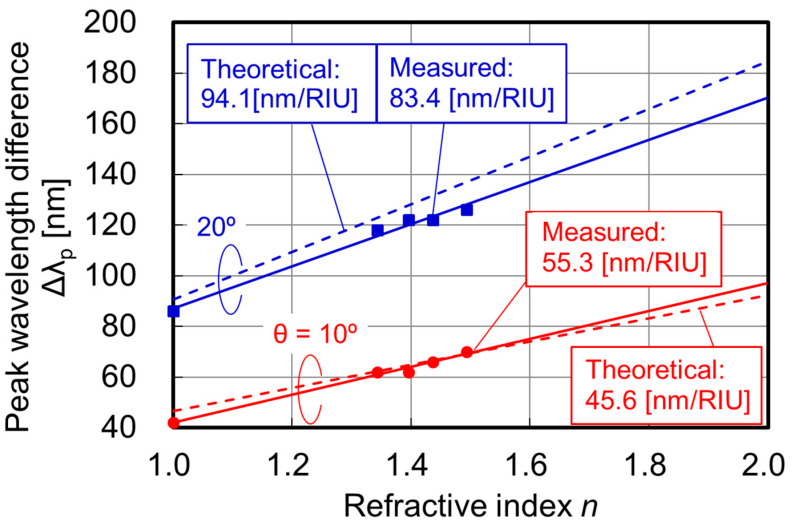
Peak wavelength difference in Figure 5 as a function of refractive index *n* with various incident angles θ. Theoretical characteristics based on Equations (2) and (3) are also shown. Sensitivity for refractive index changes can be obtained via the slope. RIU is refractive index unit.

**Figure 7 sensors-23-00568-f007:**
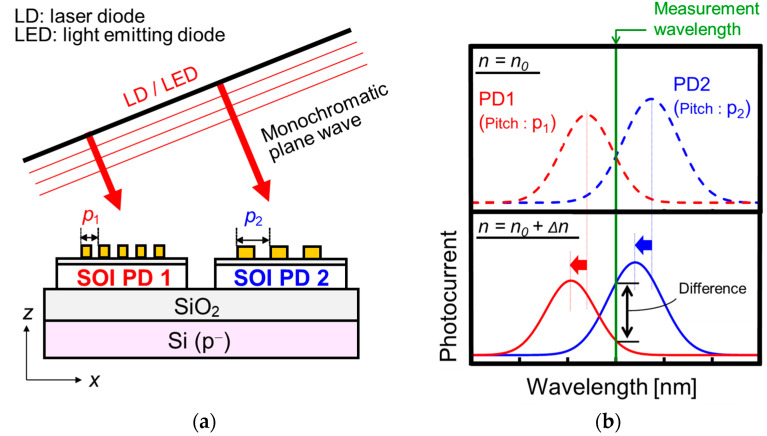
(**a**) Schematic of optical system and (**b**) operation principle for single-wavelength dual photodiode (PD) method.

**Figure 8 sensors-23-00568-f008:**
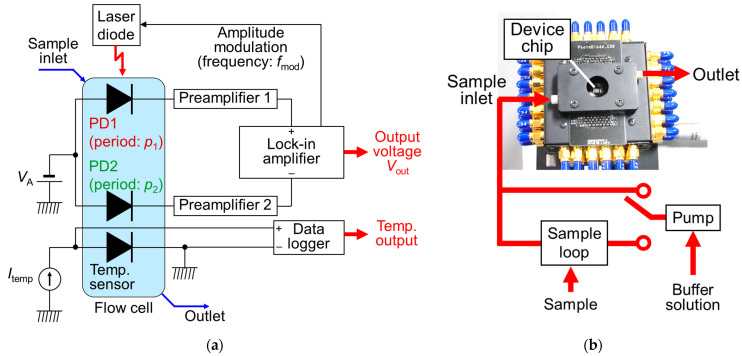
(**a**) Measurement circuit for single-wavelength dual photodiode (PD) method; (**b**) evaluation box to operate PDs with sample flow system. Packaged device chip is mounted on evaluation box, and inlet of coverage is connected to flow system.

**Figure 9 sensors-23-00568-f009:**
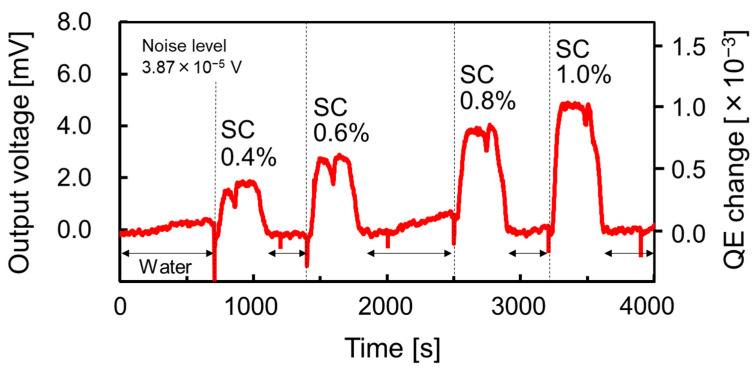
Sensorgram for aqueous solutions measured by single-wavelength dual PD method. “SC” indicates that the sucrose solution with each concentration is introduced around SP antenna as analyte. The concentrations of sucrose solutions are based on weight ratio.

**Figure 10 sensors-23-00568-f010:**
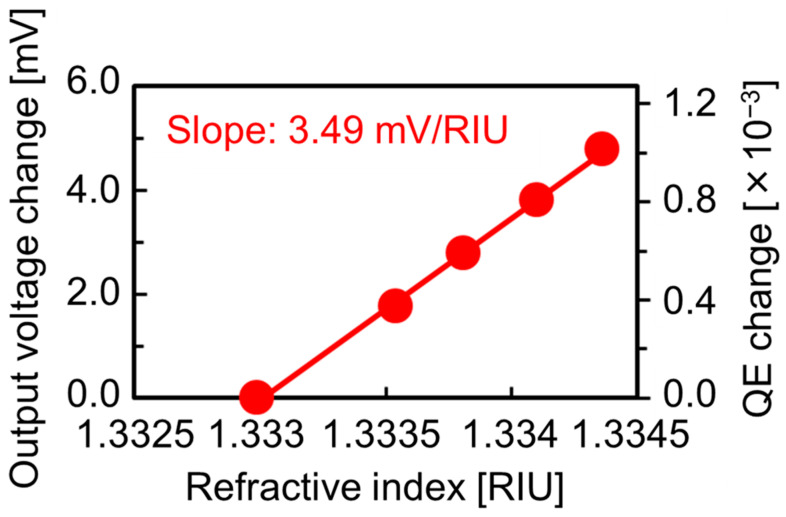
Calibration curve obtained from sensorgram in Figure 9. The differences of voltage, quantum efficiency (QE), and refractive index are taken from those of water.

**Figure 11 sensors-23-00568-f011:**
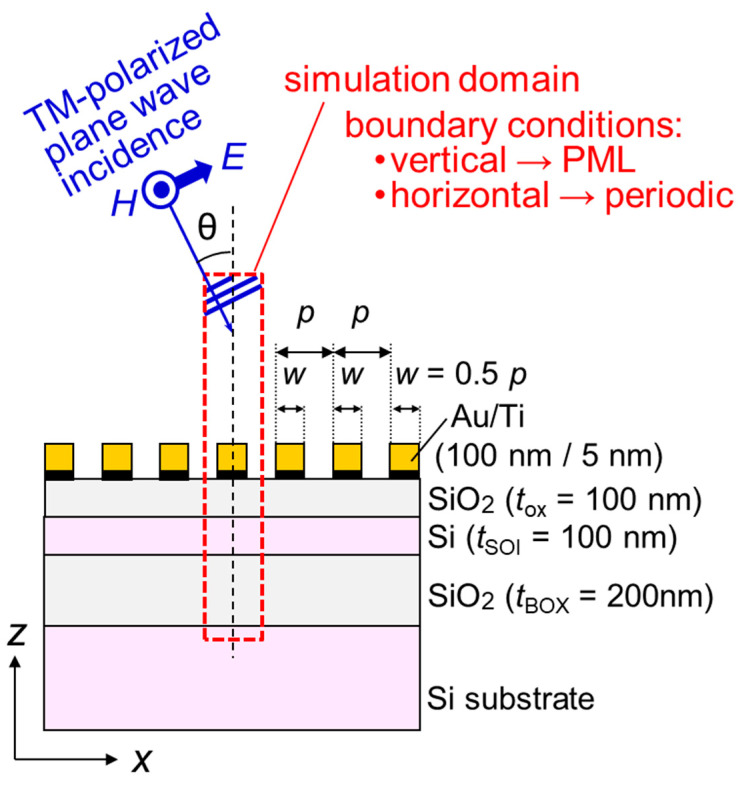
Simulated structure and conditions in finite-difference time-domain (FDTD) method for estimating quantum efficiency (QE) of SOI PD with SP antenna. PML represents perfectly matched layer, which is one of the absorbing boundary conditions used in FDTD method.

**Figure 12 sensors-23-00568-f012:**
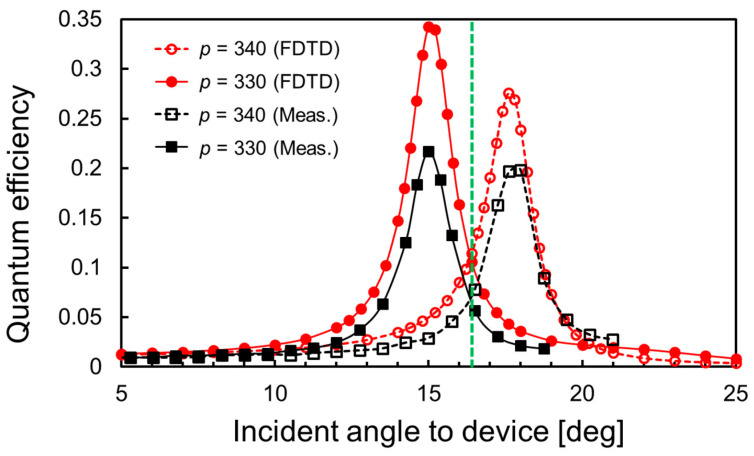
Simulated and measured quantum efficiencies (QEs) with respect to incident angle to device surface.

**Figure 13 sensors-23-00568-f013:**
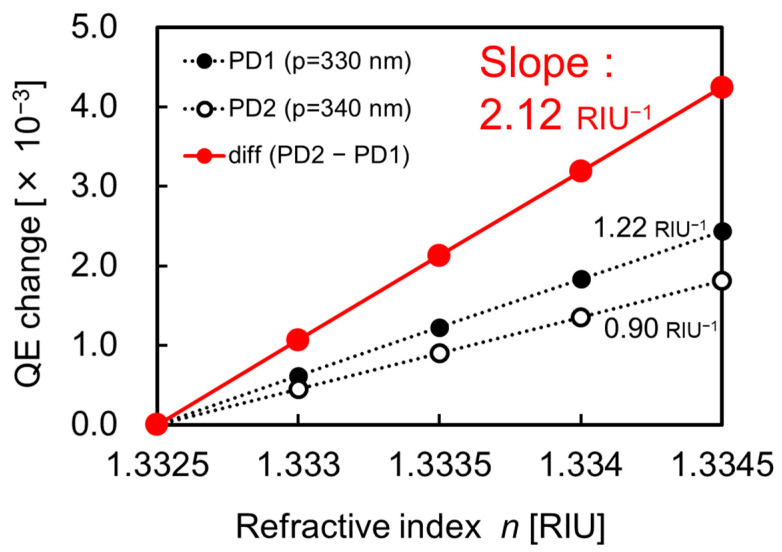
Simulated calibration curve as a function of RI around SP antenna.

**Figure 14 sensors-23-00568-f014:**
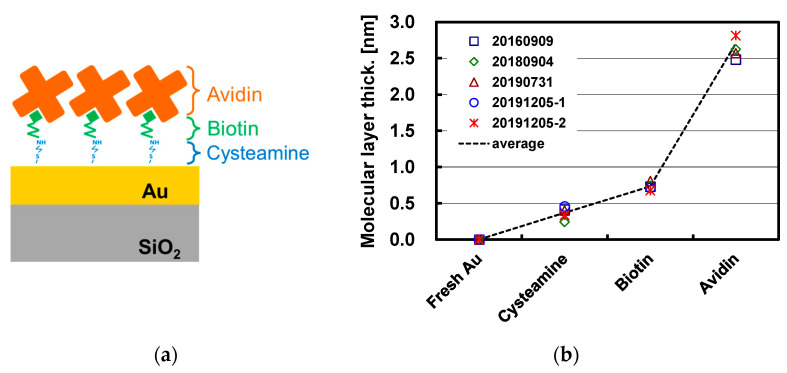
(**a**) Schematic of avidin–biotin complex and cysteamine immobilized on flat Au surface; (**b**) molecular layer thickness measured by ellipsometry with 632.7 nm light and the assumed layer refractive index of 1.5.

**Figure 15 sensors-23-00568-f015:**
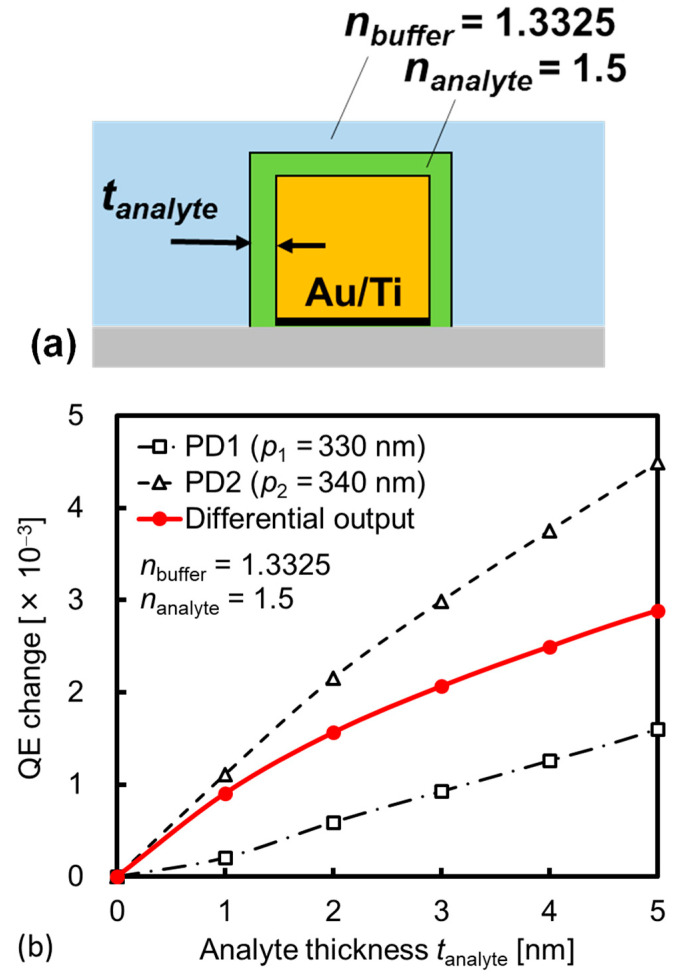
(**a**) Schematic of homogenously deposited layer on the surface of gold (Au) line of SP antenna for FDTD estimation, and (**b**) simulated QE change as a function of analyte thickness.

## Data Availability

The data that support the findings of this study are available from the corresponding author (H.S.) upon reasonable request.

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
