# Peer review of "Refractive Index Measurement Using SOI Photodiode with SP Antenna toward SOI CMOS-Compatible Integrated Optical Biosensor"

_sensors, 2023, doi:10.3390/s23020568_

Round 1

Reviewer 1 Report

Please find attachment.

Author Response

Dear Reviewer,

Thank you very much for reviewing our manuscript.

Your continued guidance would be very much appreciated.

Sincerely yours,

Hiroaki Satoh.

Reviewer 2 Report

This paper proposes a new optical biosensor composed of silicon-on-insulator (SOI) p-n 12 junction photodiode (PD) with surface plasmon (SP) antenna. I think the work is very interesting and this manuscript could be accepted.

Author Response

Dear Reviewer,

Thank you very much for reviewing our manuscript.

As the reviewer pointed out, English quality of our original manuscript was bad.

We have asked MDPI English editing service to edit the English sentences in our original manuscript.

The revised manuscript after English editing is now submitted.

Your continued guidance would be very much appreciated.

Sincerely yours,

Hiroaki Satoh.

Reviewer 3 Report

The authors propose SOI PDs with surface plasmon antenna as new biosensors.  They provide the comprehesive studies on the design, fabrication and RI measurements. The data are solid. My minor comment is that there are too many figures, especially a lot of schematics and illustrations, and many of these should be re-organized and combined with others to make the result presenations more concise. In addition, the image resolution of Figure 5 needs to be improved.
